# Perceptions of Light Pollution and its Impacts: Results of an Irish Citizen Science Survey

**DOI:** 10.3390/ijerph17155628

**Published:** 2020-08-04

**Authors:** Andrew N. Coogan, Michael Cleary-Gaffney, Megan Finnegan, Georgia McMillan, Ainhoa González, Brian Espey

**Affiliations:** 1Department of Psychology, Maynooth University, National University of Ireland, Maynooth, Ireland; Michael.ClearyGaffney@mu.ie (M.C.-G.); megan.finnegan.2017@mumail.ie (M.F.); 2Mayo Dark Skies, County Mayo, Ireland; georgia@internationalmagic.com; 3School of Geography, University College Dublin, Dublin, Ireland; ainhoa.gonzalez@ucd.ie; 4School of Physics, Trinity College, Dublin, Ireland

**Keywords:** artificial light, wildlife, sleep, public perceptions, citizen science

## Abstract

Background: Light pollution is increasingly an area of concern for health and quality of life research. Somewhat surprisingly, there are relatively few descriptions of perceptions of light pollution in the literature. The current study examined such perceptions in a Irish sample. Methods: A survey was circulated as part of a citizen science initiative of a national newspaper; the survey included questions regarding night sky brightness and the impact of light at night on sleep and animal behaviour. Complete responses from 462 respondents were analysed. Results: Urban location was, as anticipated, associated with reported brighter night skies, and public lighting was reported as the main source of light at night for urban settings, whilst neighbours’ domestic lighting was the most commonly reported source for rural settings. Respondents from rural settings were more likely to report that light at night impinged on sleep, whilst city dwellers were more likely to report recent changes in wildlife behaviour. Conclusions: Citizen science approaches may be useful in gathering data on public perceptions of light pollution and its impacts. In the current study, this perception was strongly influenced by location, highlighting the importance of assessing experiences and attitudes across a number of geographical settings.

## 1. Introduction

Man-made light pollution is an area of increasing concern from a sustainability, ecological and health perspective [1]. A recent global survey of light pollution reveals that at least 80% of the world is exposed to significant levels of artificial illumination at night [2]. Health and ecological concerns centre on the potential of light at night to act physiologically to disrupt homeostatic and behavioural control systems, such as the circadian clock that regulates daily rhythms in activity, physiology and sleep [3]. Circadian rhythms and factors that disrupt them, such as man-made lighting, are recognized as important intrinsic and environmental determinants of health [4]. A number of studies have implicated man-made artificial light at night (ALAN) with health concerns such as increased risk of hormone-dependent cancers [5] and mood disorders [6]. ALAN has also been associated with changes in wildlife behaviour due to light-induced changes in circadian phases, leading to alterations of timing of rest/activity cycles, or direct actions of light on behavioural cycles independent of circadian effects [7]. Recent evidence has indicated that, for humans, the physiological response to nocturnal light may show very high inter-individual variability, with some people responsive to very low illuminance levels [8].

Aside from the potential for direct physiological effects, the subjective perception of artificial light at night may also impact on quality of life and health-related behaviours, and may be shaped by psychological processes, such as social amplification [9]. Further, sleep disorders such as insomnia, or even subclinical poor quality sleep, have been associated with sleep attentional biases, wherein greater attention is drawn to sleep-salient factors in the environment [10]. As such, the presence of environmental light pollution has the potential to differentially and detrimentally impact on those with already poor quality sleep or sleep disorders; therefore, subjective perceptions of light pollution may shape how impactful it is for individuals’ sleep health. To date, there are surprisingly few reports in the literature on the subjective perception of ALAN and its impacts. One such study from Finland reported that light pollution was considered a nuisance for outdoors recreation, with over half of respondents reporting that light pollution reduced the overall quality of life of their neighbourhoods [11].

In this study, we report the results of a citizen science survey of experiences of light pollution in Ireland, and examine geographic and demographic features that may influence such perceptions. This study addresses an important gap in the literature regarding public perceptions of the prevalence and intrusiveness of artificial light at night.

## 2. Materials and Methods

Between March and June 2018, a brief 12 item questionnaire on light pollution was circulated via the citizen science initiative at “The Irish Times”, a national newspaper with a broad circulation (https://www.irishtimes.com/news/science/citizen-science/help-scientists-understand-the-influence-of-light-on-the-environment-1.3416898). As such, the sampling method applied was convenience sampling, and generalisability of findings from this sample was not assumed. The items on the survey asked about the nature of the home location, age, gender, a question about sky brightness at night, a question about the main source of man-made light, and five questions scored on a 7-point Likert-like scale (from “strongly disagree” to “strongly agree”) relating to perceptions of recent increase in light at night, the impact of light at night on sleep, changes in the timing of bird song, changes in the night time behaviour of animals and changes in the number of bats seen (Appendix A). The questionnaire was developed collaboratively by the authors to reflect their combined interests in light pollution, sleep health and ecology, and was designed to be appropriate for a citizen science approach. Data were fully anonymised at the point of collection, and geolocation data were not collected.

For data analysis, responses that indicated “no opinion/not applicable” were removed and Likert-like responses on 7-point scales (1 = strongly disagree; 7 = strongly agree) were analysed as ordinal data using non-parametric tests. As appropriate, pairwise comparisons between groups were conducted with Bonferroni-adjusted two-sided Mann–Whitney U scores. Correlational analysis was conducted using Spearman’s Rho for ordinal variables. Associations between categorical responses were tested with Pearson’s chi-square test. *p* < 0.05 was interpreted as indicating a statistically significant effect. All data were analysed in SPSS (IBM Corporation, Armonk, NY, USA). The statistical approach employed is exploratory, and not hypothesis testing.

## 3. Results

A total of 464 respondents completed the survey; brief demographics of the study sample are presented in Table 1.

Only 7.6% of respondents reported that the sky in their locale was completely dark, with 33.8% reporting regular visibility of the Milky Way; 38.5% reported visibility of only a few stars and not of the Milky Way, and 18.2% reported visibility only of the moon and the brighter planets (Figure 1). There was a significant large effect of location on the self-reported darkness of the night sky, with urban location being associated with subjectively brighter skies (Pearson’s chi-square = 314, df = 16, *p* < 0.001; Figure 1A). With regards to the brightest light source near to the residence, 61.9% that it was public lighting, 18.6% reported that this was their own domestic lighting, 12.8% that it was their neighbours’ lighting, 5.2% that it was commercial lighting, and 1.5% that it was passing traffic. There was a strong association of location with the source of lighting (Pearson’s chi-square = 190, df = 16, *p* < 0.001; Figure 1B), with own domestic lighting and neighbours’ lighting being important sources only in rural settings, and public lighting being the most important reported source across all settings. No statistically significant associations were found between gender or age group and the sources of light near the residence.

For items scored on a Likert-like scale relating to perceptions of possible impacts of light pollution, there was an overall neutral response to the item “The level of lighting near my home at night has increased over the past three years” (median response = 4); a disagreement with the statement “If light enters my bedroom at night it does not affect my sleep” (median response = 2); and neutral responses to “Birds sing at night (the dawn chorus starts earlier than it used to)” (median response = 4), “The natural night-time behaviour of insects/bats/foxes, etc., remains the same as in previous years” (median response = 4) and “The number of bats I see has increased recently” (median response = 3). We then examined these ratings across three groups for location (i.e., rural, town and city). There was no effect of location on ratings of the item “The level of lighting near my home at night has increased over the past three years” (Kruskal–Wallis H = 2.65, *p* = 0.266; Figure 2A). For the item “‘If light enters my bedroom at night it does not affect my sleep”, there was an effect of location (Kruskal–Wallis H = 7.74, *p* = 0.021; Figure 2B), with city dwellers endorsing this statement more strongly than rural dwellers. For the item “‘Birds sing at night (the dawn chorus starts earlier than it used to)”, there was a significant effect of location (Kruskal–Wallis H = 44.3, *p* < 0.001; Figure 2), with city inhabitant endorsing this statement most strongly. Likewise, there were statistically significant effects of location for the item “The natural night-time behaviour of insects/bats/foxes, etc., remains the same as in previous years” (Kruskal–Wallis H = 44.3, *p* < 0.001; Figure 2D, city inhabitants endorse this item the least) and a marginal effect of location for the item “‘The number of bats I see has increased recently” (Kruskal–Wallis H = 6.1, *p* = 0.049; Figure 2E, city dwellers endorse this item the least).

When age group was examined as an independent variable, there were significant effects of age on the item “The natural night-time behaviour of insects/bats/foxes, etc., remains the same as in previous years” (Kruskal–Wallis H = 9.3, *p* < 0.01; those over 55 endorse this statement most strongly) and for the item “The number of bats I see has increased recently” (Kruskal–Wallis H = 7.6, *p* < 0.023; those over 55 endorse this statement most strongly), but not on other items.

Examining inter-relatedness of the above items through simple linear regression, there are a number of statistically significant weak-to-moderate relationships (Table 2). Most notably, there is a moderate inverse relationship between “The natural night-time behaviour of insects/bats/foxes, etc., remains the same as in previous years” and “The number of bats I see has increased recently” (r = −0.304). There is also a positive relationship between “The level of lighting near my home at night has increased over the past three years” and “‘Birds sing at night (the dawn chorus starts earlier than it used to)” (r = 0.251). There were no strong associations between any of the items.

Finally, we assessed interest in night events by asking “Which of the following statements best describes your attitude towards night-themed events (such as dark sky festivals, night walks/runs, etc.)?”. Of those that expressed an opinion, 24.2% expressed a negative attitude to such events, 55.1% a neutral attitude and 20.8% a positive attitude. There were no differences across these three groups on any of the Likert-like items relating to light at night impacts, and chi-square analysis indicates that those that express an interest in night events were not more likely to live in city/town/rural settings (*p* = 0.16), but older respondents (55 or older) were more likely to have a positive attitude towards night events (*p* = 0.006).

## 4. Discussion

Citizen science approaches have previously been deployed successfully in the measurement of night sky illuminance [12], speaking to its utility for exploring other aspects of light pollution. There is a clear evidence gap in the literature around perceptions of, and beliefs about, ALAN and its impacts. A previous survey of >2000 Finnish respondents reported that light pollution was perceived to decrease the recreational amenity of outdoor spaces, that experience of dark skies was not common, that public lighting was the most commonly identified source of light pollution and that commercial lighting was the most annoying source of light at night in a predominantly urban and educated sample with a high level of interest in astronomy [11]. Our current results echo some of these findings, in that no city dwelling respondents report experiencing completely dark skies, and that public lighting was identified as the main source of light pollution for city and town dwellers, and was an important source alongside own/neighbours’ domestic lighting for rural respondents. However, our current results also do not present evidence for a perceived recent increase in the level of ALAN in rural, town or city settings. This may reflect the limited level of switches to high-illuminance LED public lighting in the past three years in Ireland and may be subject to change as such lighting becomes more prevalent.

Regarding the perceived impacts of ALAN, we find evidence that respondents endorse that light entering the bedroom impacts on sleep (a finding mostly strongly reported in rural dwellers), a finding that corresponds to finds that ALAN is associated with poorer sleep and other health outcomes in older Japanese adults [13,14], and suggests that in Ireland light pollution may be an important environmental factor to consider for sleep health. However, it should be noted that it is not presently clear whether environmental light pollution is of sufficient intensity at the incident level of the retina to produce physiological effects, or whether sleep-health impacts of ALAN may be primarily mediated through psychological mechanisms such as sleep attentional biases [1,10].

The perceived effects of light at night on animal behaviour appears to be most pronounced in city setting, findings that accord with recent reports on light pollution effects on directly observed animal behaviour [15], and is most likely reflective of greater light pollution associated with urbanisation [2]. Given that the level of exposure of wildlife to light pollution may be significantly greater than humans’ (who will mostly be indoors during the night; [1]), the magnitude of direct behavioural effects of ALAN on wildlife behaviour may be greater than that on human behaviour. Another possible link between urban setting and perceived alterations in wildlife behaviour is noise pollution, which may interact with light pollution in altering bird behaviour [16]. Perceptions of the impacts of light pollution did not vary according to interests in night-themed events, suggesting that the reported effects are not a result of general increased awareness amongst respondents with particular interests in dark skies issues.

There are a number of important caveats for the current study. Firstly, the sample is unlikely to be representative of the Irish population; rather, the sample was self-selected from those responding to the survey article in the Irish “newspaper of record”, and as such is likely to be biased towards those with pre-existing interests in ALAN, sleep and/or ecology. Second, given the nature of the survey, we did not collect either direct measures of light at night, nor other detailed demographics; nor did we collect objective measures of sleep or wildlife behaviour; future work might usefully address the associations between such subjective reports and objective records of behaviour. Thirdly, perceptions of ALAN may be shaped by other psychological constructs and beliefs; for example, such perceptions may be influenced by social amplification [5]. Fourthly, given the strictures of a citizen science project, we did not use established psychometric scales for assessing sleep or other domains, given the imperative for brevity in the survey design. Fifthly, future work might usefully address whether chronotype (preference towards earlier or later timing of sleep/wake) influence perception of ALAN. Overall, the current study indicates that experience of man-made light at night is common in Ireland, but varies by geographic location and age, with older age and city location associated with the greatest perceived effects on sleep and animal behaviour.

## 5. Conclusions

This study indicates that citizen science approaches may be useful in gaining insight into public perceptions of man-made lighting. Further, this study indicates that there are differences in perceptions of the presence and impact of light pollution across rural and urban settings and that perceptions of some light-associated effects (e.g., on wildlife behaviour) are more commonly reported in older respondents. Future studies will be needed to assess the relationship between subjective self-reported levels of light at night and objectively assessed levels in order to replicate the current findings and to broaden the scope of factors examined.

## Figures and Tables

**Figure 1 ijerph-17-05628-f001:**
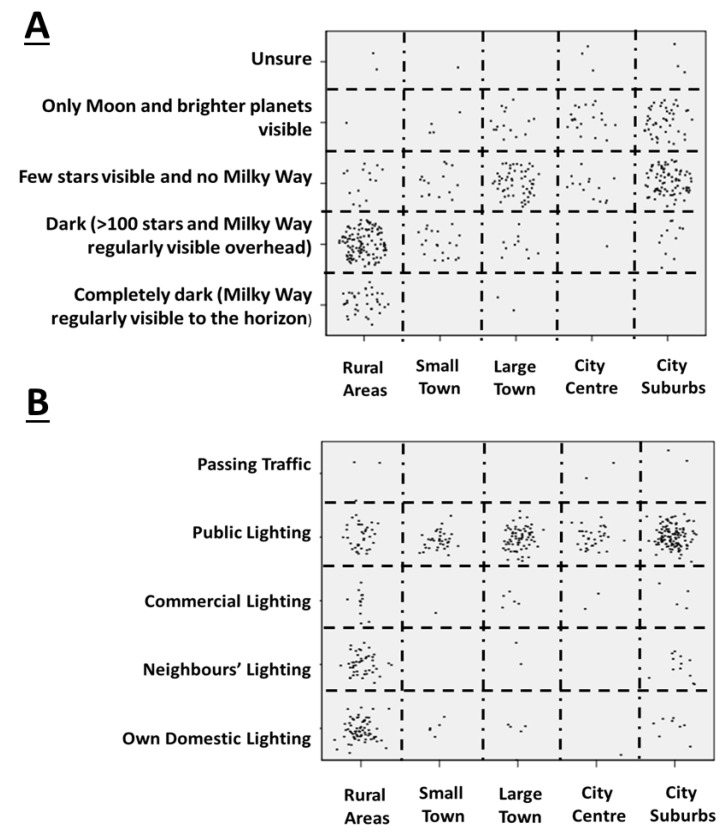
Number of responses for items on (**A**) dark skies and (**B**) strongest light source near home by self-reported residence location.

**Figure 2 ijerph-17-05628-f002:**
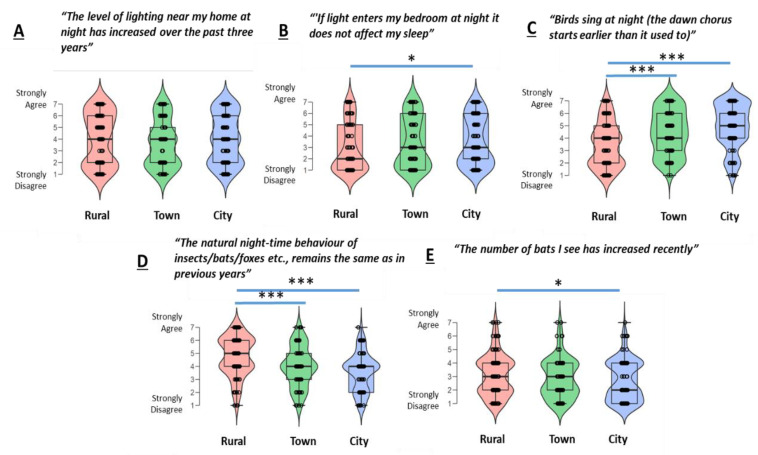
Box and violin plots for responses on Likert-like items relating to individual questions on perceived impacts of light pollution (**A**–**E**) compared across three groups of self-reported residence location. ***** indicates *p* < 0.05 for Bonferroni-adjusted pairwise comparison by Mann–Whitney U test; ******* denotes *p* < 0.005.

**Table 1 ijerph-17-05628-t001:** Demographics of the Study Sample.

Female *n* = 230; 49%	Male *n* = 234; 51%
Age (years)	Residential Location
18−24: 10.8%	Rural: 35.3%
25−34: 26.9%	Small Town: 8.6%
35−44: 22.4%	Large Town: 17.5%
45−54: 15.1%	City Centre: 8.2%
55−64: 7.1%	City Suburb: 30.4%
65+: 7.1%	

**Table 2 ijerph-17-05628-t002:** Inter-item correlation matrix for Likert-like items on perceived impacts of light pollution.

		Night Behaviour Animals	Light Increased Near Home	Light and Sleep	Birds Sing at Night
Night Behaviour Animals	Spearman’s rho	-	-	-	-
	*p*-value	-	-	-	-
Light Increased near Home	Spearman’s rho	−0.143	-	-	-
	*p*-value	0.006	-	-	-
Light and Sleep	Spearman’s rho	0.065	-0.132	-	-
	*p*-value	0.217	0.007	-	-
Birds Sing at Night	Spearman’s rho	−0.303	0.251	0.032	-
	*p*-value	<0.001	<0.001	0.519	-
Number of Bats	Spearman’s rho	0.099	0.169	0.028	0.162
	*p*-value	0.067	0.001	0.584	0.002

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
