# Peer review of "Perceptions of Light Pollution and its Impacts: Results of an Irish Citizen Science Survey"

_ijerph, 2020, doi:10.3390/ijerph17155628_

Round 1
Reviewer 1 Report
- The article should state what light sources were located around the homes of people participating in the survey.
- It could also be presented in this article whether there is a correlation between the type of light source and the informations of respondents.
- The presentation of the data on Figure 1. is not legible. The readability of this drawing should be improved
Author Response
Reviewer#1
- The article should state what light sources were located around the homes of people participating in the survey.
Authors’ reply: There was a specific item on the questionnaire asking participants to indicate the most significant light-sources in the vicinity of their residences; this data is summarised in Figure 1B. We did not ask about sources of indoor lighting, as we felt this is a separate (but very important) area of investigation.
- It could also be presented in this article whether there is a correlation between the type of light source and the information of respondents.
Authors’ reply: we have now analysed the sources of light near the residences against both gender and age group, and report no association: (line 103: No statistically significant associations were found between gender or age group and the sources of light near the residence.)
- The presentation of the data on Figure 1. is not legible. The readability of this drawing should be improved.
Authors’ reply: We have amended Figure 1 to improve its readability.
Reviewer 2 Report
The article concerns the current and important problem of a practical nature. The purpose of research is a consequence of the thesis adopted by the authors. The introduction to the topic presented seems to be too short. Adding a few or several scientific literature items with a short comment would undoubtedly improve the work. In future research, it is worth considering the chronotype of the surveyed people, that is an individual feature of people, describing the preferences for choosing the morning or evening time of activity.
It is worth improving the quality of figure 1.
Comments and observations presented in this review have an ancillary nature and do not decrease the scientific value of the presented article.
Author Response
- The article concerns the current and important problem of a practical nature. The purpose of research is a consequence of the thesis adopted by the authors. The introduction to the topic presented seems to be too short. Adding a few or several scientific literature items with a short comment would undoubtedly improve the work. In future research, it is worth considering the chronotype of the surveyed people, that is an individual feature of people, describing the preferences for choosing the morning or evening time of activity.
Authors’ reply: We have now expanded the introduction to elaborate on possible physiological and psychological mechanisms through which ALAN might impact on behaviour. As regards to chronotype, we agree that this is an important construct to consider (and we are indeed doing so currently in a separate study) and highlight this as an area for future consideration: (line 204: Fifthly, future work might usefully address whether chronotype (preference towards earlier or later timing of sleep/wake) influence perception of ALAN.)
- It is worth improving the quality of Figure 1.
Authors’ reply: We have amended Figure 1 to improve its readability.
Reviewer 3 Report
This work could initially raise some expectations as a case study applied to a specific territory. However, analyzing it in-depth, it presents clear deficiencies that should be corrected before being published as a scientific article. We go into detail some specific issues
Abstract:
Abstract it should be avoided to number 1. (Background). (2) Methods, etc ... just describe briefly the object of study, its justification and results. It is recommended to rewrite the Abstract again more clearly and appropriately. There is talk of a sample number N but it must be somewhat more explicit. It´s very confuse.
1.Introduction
A deep revision of the Introduction is recommended looking for recent works with updated bibliographic references. We do not agree with the expression that there are few works related to the object of study
line 40 "To date, there are surprisingly few reports in the literature on the perceptions of man-made light-at-night and its impacts".
Line 46 “important gap in the literature regarding public perceptions of the prevalence and intrusiveness of artificial light-at-night.”
There are many reviews of complete papers on this topic. It is necessary to rewrite the Introduction in a more scientific way and supported by works and studies on the subject.
A few examples:
- Light Pollution, Circadian Photoreception, and Melatonin in Vertebrates DOI: 3390/su11226400
- The impact of artificial light at night on human and ecosystem health: a systematic literature review DOI: 10.1007/s10980-020-01053-1
- Streetlights Disrupt Night-Time Sleep in Urban Black Swans DOI: 3389/fevo.2020.00131
- Working with Inadequate Tools: Legislative Shortcomings in Protection against Ecological Effects of Artificial Light at Night DOI: 3390/su12062551
- Materials and Methods.
This section is very poor. The Methodology must be explained in detail, not referring the Methodology to a publication that is not even scientific. If questionnaires have been prepared, you must indicate how they were done, what sample and the geographic area has been sampled and represent everything in the form of tables that allow the study to be analyzed. This section should be written again in much more detail and much more clearly. It is confusing and has no scientific weight methodologically speaking
Line 51. Avoid referencing WEB pages within the text. All references must be numbered and referenced at the end of the text in their References section.
- Results
The results are also very confusing. Tables of results that really show the same is required. The figures are practically illegible and of very low quality, making it impossible to observe anything. All the statistical aspects of the study must be previously explained in the Methodology section, not in the Results section. We did not observe any bibliographic reference to similar studies that justify the applied methodology or that relate the results obtained with other results from other studies.
- Discussion
The discussion is poor and requires a further review of the literature related to the subject. It should also support the study biased by age range, geographical situation, and social aspects related to the sample to give greater solvency to the discussion of results. It is recommended to completely rewrite
- Conclusions:
The conclusions are excessively concise and do not provide scientific novelty related to the study carried out. It is recommended to rewrite them again relating the results with the objectives that must be exposed in the methodological part. And clearly show the scientific contribution of the study based on the results obtained and future lines of research.
- References
References are few, low impact and out of date. A much more rigorous bibliographic review work must be carried out so that the work has real and current scientific solvency.
Author Response
Reviewer#3:
- Abstract it should be avoided to number 1. (Background). (2) Methods, etc ... just describe briefly the object of study, its justification and results. It is recommended to rewrite the Abstract again more clearly and appropriately. There is talk of a sample number N but it must be somewhat more explicit. It´s very confuse.
Authors’ reply: We have removed the numbering in the abstract. It is unclear what aspects of the abstract were not clear to the reviewer. Further, it is not clear to us how stating a clear number of participants (N=462) could be any more explicit.
- A deep revision of the Introduction is recommended looking for recent works with updated bibliographic references. We do not agree with the expression that there are few works related to the object of study
line 40 "To date, there are surprisingly few reports in the literature on the perceptions of man-made light-at-night and its impacts".
Line 46 “important gap in the literature regarding public perceptions of the prevalence and intrusiveness of artificial light-at-night.”
There are many reviews of complete papers on this topic. It is necessary to rewrite the Introduction in a more scientific way and supported by works and studies on the subject.
A few examples:
Light Pollution, Circadian Photoreception, and Melatonin in Vertebrates DOI: 3390/su11226400
The impact of artificial light at night on human and ecosystem health: a systematic literature review DOI: 10.1007/s10980-020-01053-1
Streetlights Disrupt Night-Time Sleep in Urban Black Swans DOI: 3389/fevo.2020.00131
Working with Inadequate Tools: Legislative Shortcomings in Protection against Ecological Effects of Artificial Light at Night DOI: 3390/su12062551
Authors’ reply: The title of our study is “Perceptions of Light Pollution and its Impacts: Results of an Irish Citizen Science Survey.” As such, the study assessed subjective perceptions of light pollution, an area in which there is indeed very little work published. None of the articles mentioned by the reviewer here pertain to subjective perceptions of light pollution; rather they address the area of physiological and behavioural responses to ALAN, an area in which there is indeed a more developed literature. We have expanded our introduction to highlight some of these areas, but do not elaborate greatly as 1) this area is not the primary objective of the study and 2) this is a brief communication.
- This section is very poor. The Methodology must be explained in detail, not referring the Methodology to a publication that is not even scientific. If questionnaires have been prepared, you must indicate how they were done, what sample and the geographic area has been sampled and represent everything in the form of tables that allow the study to be analyzed. This section should be written again in much more detail and much more clearly. It is confusing and has no scientific weight methodologically speaking
Line 51. Avoid referencing WEB pages within the text. All references must be numbered and referenced at the end of the text in their References section.
Authors’ reply: Again, it is not fully clear to us what aspects of the methods section the reviewer finds unclear, as they are not specific. We have amended the methods sections to contain a statement on the development of the questionnaire (Line 73: The questionnaire was developed collaboratively by the authors to reflect their interests in light pollution, sleep health and ecology, and was designed to be appropriate for a Citizen Science approach) and to further clarify the sampling method (line 66: As such, the sampling method applied was convenience sampling, and generalizability of findings from this sample was not assumed). The basic demographics collected are now represented in Table 1. We include the weblink to the original survey and newspaper article as this provides the reader with full access to the context of the original presentation of the survey, maximising scientific transparency.
- The results are also very confusing. Tables of results that really show the same is required. The figures are practically illegible and of very low quality, making it impossible to observe anything. All the statistical aspects of the study must be previously explained in the Methodology section, not in the Results section. We did not observe any bibliographic reference to similar studies that justify the applied methodology or that relate the results obtained with other results from other studies.
Authors’ reply: Again, it is not fully clear to us what aspects of the results presentation the reviewer finds unclear, but we have endeavoured to improve this section. We have included now some demographic information in Table 1. We have amended Figure 1 to improve its readability. We believe the methods sections contains sufficient information on the statistical approaches employed. According to the Journal guidelines, the results section should not contain discussion of the presented data with regards to the extant literature.
- The discussion is poor and requires a further review of the literature related to the subject. It should also support the study biased by age range, geographical situation, and social aspects related to the sample to give greater solvency to the discussion of results. It is recommended to completely rewrite.
Authors’ reply: We have expanded the discussion to further elaborate on issues such as the roles of sleep attentional biases in perception of light pollution. We have further expanded on the limitations of the study, and caveats to the interpretation of the results (lines 194-208). If there are further papers on subjective perceptions of light pollution that this reviewer is aware of that we have not cited or discussed, we would be pleased to include these in any further revision.
- Conclusions: The conclusions are excessively concise and do not provide scientific novelty related to the study carried out. It is recommended to rewrite them again relating the results with the objectives that must be exposed in the methodological part. And clearly show the scientific contribution of the study based on the results obtained and future lines of research.
Authors’ reply: It is not clear to us what “excessively concise” is meant to convey as a criticism of the conclusions paragraph. We had already stated in the original version how the current results may inform future work in this area (line 204: Future studies will be needed to assess the relationship between subjective self-reported levels of light-at-night and objectively assessed levels, to replicate the current findings and to broaden the scope of factors examined.)
Round 2
Reviewer 3 Report
We respect the authors 'efforts to improve the work, but we consider the authors' responses and the changes made to be insufficient. Eg, Figure 1 for example despite the change is practically not legible.
We would only recommend the editors to publish this work as a technical note or brief communication. Never as a scientific article, because it does not have the necessary quality